# Current Opinion on the Use of c-Fos in Neuroscience

**Sandra Yasbeth Lara Aparicio** [1,2,*] , **Ángel de Jesús Laureani Fierro** [3], **Gonzalo Emiliano Aranda Abreu** [1],
**Rebeca Toledo Cárdenas** [1], **Luis Isauro García Hernández** [1], **Genaro Alfonso Coria Ávila** [1], **Fausto Rojas Durán** [1],
**María Elena Hernández Aguilar** [1], **Jorge Manzo Denes** [1], **Lizbeth Donají Chi-Castañeda** [1]
**and César Antonio Pérez Estudillo** [1]

1    Instituto en Investigaciones Cerebrales, Universidad Veracruzana, Xalapa de Enríquez,
     Veracruz C.P. 91190, Mexico
2    Laboratorio de Neurofisiología, Instituto de Investigaciones Cerebrales, Universidad Veracruzana, Av. Luis
     Castelazo S/N, Col. Industrial Las Ánimas, Xalapa de Enríquez, Veracruz C.P. 91190, Mexico
3    Doctorado en Investigaciones Cerebrales, Universidad Veracruzana, Xalapa, Veracruz C.P. 91190, Mexico
*    Correspondence: hikari_yagami17@hotmail.com

**Abstract:** For years, the biochemical processes that are triggered by harmful and non-harmful stimuli at the central nervous system level have been extensively studied by the scientific community through numerous techniques and animal models. For example, one of these techniques is the use of immediate expression genes, which is a useful, accessible, and reliable method for observing and quantifying cell activation. It has been shown that both the *c-fos* gene and its protein c-Fos have rapid activation after stimulus, with the length of time that they remain active depending on the type of stimulus and the activation time depending on the stimulus and the structure studied. Fos requires the participation of other genes (such as *c-jun*) for its expression (during hetero-dimer forming). c-Fos dimerizes with c-Jun protein to form factor AP-1, which promotes the transcription of various genes. The production and removal of c-Fos is part of cellular homeostasis, but its overexpression results in increased cell proliferation. Although Fos has been used as a marker of cellular activity since the 1990s, which molecular mechanism participates in the regulation of the expression of this protein is still unknown because the gene and the protein are not specific to neurons or glial cells. For these reasons, this work has the objective of gathering information about this protein and its use in neuroscience.

**Keywords:** c-Fos; early gene; brain activity; cell activation; transcription; stimuli; animal models

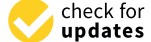



## 1. Introduction

Immediate and early expression genes (IEGs) comprise a group of genes that are activated immediately in response to a specific and recent stimulus and that do not require the synthesis of new proteins, as in the case of c-Fos. The *c-fos* gene encodes a nuclear protein called Fos (though its activation is changed by the addition of a phosphate group mediated by kinases). Members of the c-Fos family dimerize with the c-Jun protein to form the transcription factor AP-1, which activates the transcription of a range of numerous and diverse genes that are related to cell differentiation and proliferation, learning, motor control, and cognition [1]. The *c-fos* gene and the c-Fos protein are markers of cellular activity (for both neurons and glia) and easily expressed after a wide range of stimuli (both harmful and non-harmful), which makes them useful in various animal models and when used in conjunction with other labeling techniques. Currently, c-Fos immunohistochemistry is still widely used, allowing active cells to be identified through a representative black nuclear label and then quantified. The methodology is relatively simple and standardized compared to other current techniques; however, many researchers have questioned its use due to its limitations. Thus, in the present article we provide details about the foundations of using *c-fos* and c-Fos as markers of cell activation in neuroscience and discuss the

associated advantages and disadvantages. Although this work seeks to delve into Fos, there is still much to investigate, detail, and expand.

## 2. Methodology: Selection Criteria

In the present review, scientific articles related to Fos, both the gene and the protein, were analyzed. The PRISMA-P recommendations were followed for the systematic review. For this purpose, the PubMed, GoogleScholar, and SciELO electronic databases were searched. The exploration identified studies using various animal models. The following keywords were used in Spanish and English: c-Fos, early gene, brain activity, cell activation, transcription, stimuli. A total of 3245 articles were registered between 1984 and 2021.

The selection criteria for these papers were as follows:

Study population: determined according to the total number of papers containing the keywords.

Inclusion criteria: papers that were related and which had the keywords of this review.

Exclusion criteria: papers that were not related to the research topic, were unrelated in title, or were duplicates.

Sample: the sample was determined according to the papers that met the inclusion criteria ($n$ = "x" number of papers).

The procedure for identifying the papers that fulfilled the eligibility criteria and which were included, described, and analyzed in this review was as follows:

(a). Records identified in GoogleScholar ($n$ = 3000).
(b). Records identified in SciELO and Pubmed ($n$ = 245).
(c). Total papers ($n$ = 3245).
(d). Papers excluded based on title/being duplicates/being deleted ($n$ = 2888).
(e). Papers selected to assess their eligibility ($n$ = 357).
(f). Papers included in this review ($n$ = 82).

## 3. Development

### 3.1. The Expression of Immediate Early Genes (IEGs)

IEGs are rapidly and transiently activated upon cellular stimulation, forming transcription factors that modify the expression of other genes. These IEGs are induced in specific brain regions in response to behavioral tasks. The term "immediate and early expression" was initially assigned by virologists to describe exclusively viral regulatory factors transcribed de novo by host cells immediately within 2 min of viral integration. After this time was exceeded, the term "delayed early expression" was coined. Subsequently, this concept was adopted by biologists, who extended the time threshold from 2 min to 30 and 60 min, the time at which maximum peaks are reached for certain IEGs; biologists thus defined them not only by transcription time, but as genes that can be expressed without de novo protein synthesis [1,2].

IEGs are classified into two groups: (a) those that encode transcription factors that regulate the expression of other genes by binding to a specific DNA sequence [called regulatory transcription factors (RTFs)] and (b) those that encode proteins that are found throughout the cytoplasm and which go into the peri-dendritic region of cells, directly modifying cell function (called effector IEGs) [1,2]. Some examples of IEGs, and their functions, are shown in Table 1. Table 2 shows some stimuli and animal models that have been used to study some IEGs.

**Table 1.** Classification of IEGs and their functions.

| IEGs | Function |
| --- | --- |
| *Arc* (Arg3.1) | It regulates specific neurons associated with learning and memory. These functions are also associated with *c-fos* and *egr-1*. |

**Table 1.** *Cont.*

| IEGs | Function |
|---|---|
| *c-fos* *c-jun* | Cell activation (cell proliferation and differentiation) and protein transcriptions. For *c-fos*: sensory information processing, pain regulation mechanisms, long-term potentiation, neural plasticity, learning and memory, direct control of the expression of inflammatory cytokines, mediation of neuronal excitability by regulating the expression of the kainic acid receptor and GluR6, brain-derived neurotrophic factor. |
| *cyr 61* | Cell adhesion, migration, and angiogenesis. |
| *egr-1* (*tis 8*) | Functioning and development of the central nervous system, as well as the development of prostate cancer, thymic T cells. Determines the fate of hematopoietic cells. |
| *erp* or *MKP-1* (dual-specificity MAPK phosphatase) | Dephosphorylation of MAPs, essential for the regulation of cell growth. |
| *Homer1a* | Long-term potentiation, memory consolidation processes. and modification of glutamatergic signaling pathways. |
| *krox-24* | Long-term potentiation, neural plasticity, learning and memory. |
| *c-myc* | Normal cell development, cell malignancy, cell proliferation and apoptosis. |
| *nur1* *nur77* (*NGFI-B*) | Vision (dorsal lateral geniculate nuclei and visual cortex), sensory information processing (primary olfactory cortex and anterior olfactory nuclei), pain regulation mechanisms (greater raphe nuclei), auditory stimulation and stress mechanisms (facial, spinal trigeminal, medial vestibular, lateral vestibular, and dorsal cochlear nuclei). |
| *N-10* (thyroid hormone receptor) | Cell differentiation and proliferation. |
| *pip92*, *Chx1* or *ETR101* | Neuronal differentiation and cell death. |
| *tis10* (*cox-2*) | Formation of PG-G$_2$ from arachidonic acid and two oxygen molecules, which is then reduced to PGH$_2$ and, in turn, acts as a precursor to several eicosanoids, including PGE$_2$, PGF$_{2\alpha}$, PGD$_2$, prostacyclin, and thromboxane A$_2$. |
| *tis11* | Gene expression at the transcriptional level, promotes destabilization of cytokine $_m$RNAs and promotes ARE (adenylate-uridylate-rich elements)-induced $_m$RNA decay and decay. |
| Glutamate receptor 6 (GluR6), mitogen-activated proteins (MAPs), PG (prostaglandin) [3–10] | |
| *tis21* | Proliferation and differentiation of neural stem and progenitor cells (for example, in the cerebellum, hippocampus, or dentate gyrus). |
| *TSP-1* (thrombospondin-1) | Activation of receptors and signaling pathways, formation of multimolecular complexes, sequestration and inactivation of growth factors and enzymes, alterations in protein localization, proteolytic processing and internalization, and effects on receptor/ligand balance and downstream signaling (depending on their ability to bind to different ligands). |

**Table 1.** *Cont.*

| IEGs | Function |
|------|----------|
| *Stra13*<br>*zf9* | Early phase of preadipocyte cell differentiation. |
| *zif268* (*NGFI-A*) | Synaptic plasticity and long-term memory. |
| [8,9,11–13] | |

**Table 2.** Stimuli and animal models that lead to the expression of some IEGs.

| IEGs | Stimulus | Animal Model |
|------|----------|--------------|
| *Arc* (*Arg3.1*) | Learning and memory | Rats |
| *c-fos*<br>*c-jun*<br>*c-myc* | Seizures<br>Cell differentiation<br>Neural arousal<br>Electrical stimulation<br>Surgical injuries/nerve transections<br>Devascularization<br>Cerebral ischemia<br>Abstinence<br>Nociceptive and peripheral stimulation<br>Heat stress<br>Light stimulation<br>Old age | Mice<br>Rats<br>Cats<br>Monkeys<br>Humans<br>Dogs |
| *egr-1* (*tis 8*) | Phorbol esters<br>Growth factors (e.g., NGF, EGF)<br>Cellular differentiation (cardiac<br>and neuronal)<br>Electrical stimulation | *Drosophila* fruit fly<br>Rats<br>Mice |
| *egr-2* (*krox20*)<br>*cyr 61*<br>*nur77* (*NGFI-B*) | Cell differentiation<br>Growth factors | Developing mice Rats |
| *krox-24*<br>*N-10*<br>*zif268* (*NGFI-A*) | Phorbol esters<br>Growth factors (e.g., NGF)<br>Long-term object recognition and<br>memory tasks (24 h) | *Drosophila* fruit fly<br>Mice |
| [4,10,13–16] | | |

### 3.2. Fos

The mechanisms that maintain and repair DNA can be affected by hereditary or sporadic mutations that are present in all cells, leading to the substitution of amino acids in critical portions of proteins through the action of abnormal genes called oncogenes (for example, *ras*, *myc*, *fos*, and *c-fms*), which in their normal state are known as proto-oncogenes. Both oncogenes and proto-oncogenes can have a cellular or viral origin, whose homology was established in 1976 by Stehelin [17].

Fos is derived from cellular genes, referred to as *c-fos*, and from viral genes, referred to as *v-fos*, a nomenclature that applies to all other IEGs. It should be noted that to refer to the gene or its protein product, most research uses lowercase letters and italics to designate the gene and uppercase letters for the product, e.g., *c-fos* and c-Fos, respectively [18–24].

#### 3.2.1. c-Fos Gene

The term "fos" was first used to describe the oncogene encoded by the Finkel–Biskis–Jinkins murine osteogenic sarcoma virus (FBJ-MSV), thus taking the name viral oncogene (*v-fos*). Being an oncogenic-type retrovirus, FBl-MSV is derived from the recombination of retroviral sequences with cellular genetic information. These normal cellular sequences

from which *v-fos* was derived were termed the fos or *c-fos* proto-oncogene, which is located at the level of the cell nucleus in a chromosomal context. Homology between *v-fos* and *c-fos* does not occur in four regions, of which three are introns with the appropriate splice donor or acceptor site and the last represents sequences that have been deleted in the biogenesis of *v-fos*. Thus, although homologous, they have different carboxyl termini [1,15,20,25].

Basal expression of *c-fos* has been studied in several species, including mice, rats, cats, monkeys, and humans. Although its basal levels are relatively low due to the instability of its $_m$RNA, its transcription is rapid (within the first 5 min), continuing for up to 15–20 min. Accumulation of its $_m$RNA peaks between 30 and 45 min after stimulation and decreases with a short half-life of $\approx$10–20 min, which self-represses transcription by its own Fos protein (negative feedback loop). The rapid transcription of *c-fos* led to its classification as an IEG [3,15,16,26]. It is important to consider that the time for basal expression of c-Fos and *c-fos* depends on cell function (such as cell growth, differentiation, or survival) and brain areas (some do not express Fos or express it in low concentration, as shown by Herdegen et al. in 1995). On the other hand, the expression of c-Fos and *c-fos* after stimuli depends on the type (will be detailed later). For example, Fos levels after a spinal cord injury are different from those obtained after epileptic seizures, which are different from those that occur through pharmacological induction, etc. Although there is little information on studies related to the time for basal expression of *c-fos* and c-Fos, the studies found seem to indicate similar times for expression of the *c-fos* gene ($_m$RNA) and the c-Fos protein (Table 3).

**Table 3.** Expression time of c-Fos and *c-fos* (basal and after stimuli).

|  | Basal Expression * | After Stimuli Expression ** |
|---|---|---|
| $_m$RNA transcription | $\approx$first 5 min | $\approx$first 15 min |
| $_m$RNA increase | $\approx$15–20 min | $\approx$30–60 min |
| $_m$RNA peak | $\approx$30 min | $\approx$30–45 min |
| $_m$RNA half-life (metabolization) | $\approx$180 min | $\approx$10–20 min |
| c-Fos transcription | — | $\approx$20–90 min |
| c-Fos increase | — | Depends on type of stimuli |
| c-Fos peak | — | Depends on type of stimuli |
| c-Fos half-life (metabolization) | — | $\approx$2–5 h |

[15,16,26,27]

* Basal expression depends on cellular function and brain area. ** After stimuli expression depends on the type of stimuli.

Although this gene and its protein product are the most widely used markers of neuronal activation, their use has several disadvantages: (a) because both the gene and the protein can be activated by an infinite number of signaling pathways, their expression does not provide specific information about a given pathway; (b) their expression time is limited, thus providing little information about their degree of activation; (c) both the gene and the protein are not exclusive to neurons or glia since they occur in both nerve cells, thus their identification is not accurate for the type of active cell; (d) although many stimuli can activate *c-fos* and c-Fos, not all can activate both, which limits their use; and (e) it is used only to measure cell activation, it is not useful for inhibitory processes. However, the use of other techniques, such as imaging or in vivo electrophysiological recording, is time consuming, demanding, and expensive. Therefore, the use of *c-fos* and c-Fos remains one of the most widely used methods for assessing cell activation in vivo [28].

### 3.2.2. Fos Protein

The Fos family consists of four cellular proteins (c-Fos, FosB, Fra-1, and Fra-2), and its expression is related to the cell cycle. The *c-fos* gene encodes a protein called c-Fos, which has 380 amino acids and a molecular weight of 56–62 kDa. Among murine v-Fos and c-Fos, the first 332 amino acids are similar and the other 48 amino acids encode in different reading frames. Modification corresponds more to the sequence and less to the phosphorylation of serine and threonine (this part is temporary and is related to the function of the molecule). Despite this, both murine c-Fos and v-Fos are found in the cell nucleus. The Fos protein undergoes more extensive post-translational modifications than v-Fos [e.g., increase in molecular weight, acetylation, phosphorylation, (poly)ADP-ribosylation, and ubiquitination] [1,15,23,25,29].

The synthesis of c-Fos (Figure 1) follows the accumulation of its gene $_m$RNA and could be detected by immunohistochemistry (IHC) between 20 and 90 min after stimulation, although it has also been observed up to 24 h after stimulation (in the cell nucleus). This protein has a half-life of approximately 2 h [3,14–16,18,20,24,30–36]. c-Fos activation is regulated by interactions with various transcriptional regulators, including $Ca^{2+}$, cyclic adenosine monophosphate (cAMP or cAMP), or mitogen-activated protein kinases (MAPKs) [35,36].

### 3.2.3. AP-1, MAPK, and c-Fos Activation

The MAPK cascade is a cellular signaling pathway that regulates both the expression and post-translational modification of several proteins, including activator protein 1 (AP-1). AP-1 is a transcription factor that controls and regulates the transcription of genes linked to vital long-term cellular processes such as growth, differentiation, development, proliferation, transformation, migration, apoptosis, and oncogenesis [16,37,38]. This transcription factor belongs to three different protein subfamilies: Fos, Jun (c-Jun, JunB, and JunD), and Activating Transcription Factor (ATFa, ATF2, ATF3, ATF4, and B-ATF) [21,23,38].

The MAPKs present three subfamilies based on the sequence of the phosphorylation motif: JNKs (c-Jun N-terminal kinases), ERKs (extracellular signal-regulated kinases), and p38 (also known as CSBP, mHOG1, RK, and SAPK2). Specifically, the ERK subfamily is related to cell proliferation and differentiation, while JNK and p38 respond to cell stress and survival. Each subfamily is activated by MAPs, which are integrated into three consecutive elements: MAP3K (MAP kinase kinase kinase, or MEKK), MAP2K (MAP kinase kinase, or MEK), and MAPK (MAP kinase). Their sequence indicates the order in which they were consecutively activated by phosphorylation reactions [39,40].

Activation of the AP-1 pathway (and subsequent activation of IEGs) begins with an influx of calcium into the neuron through a combination of the N-methyl-D-aspartate glutamate receptor (NMDAr) and voltage-gated calcium channels (e.g., type L calcium). This entry of calcium can be caused by multiple internal and external stimuli. If we talk about external factors (including ERK), we find growth factors, cytokines, seizures (electrical or neurochemical), external stimulations (electrical, physiological, nociceptive, and metabolic), stress, sexual behavior, exposure to light, administration of various substances and chemicals, etc. [19,29,35,38,39,41].

Calcium can flow into the neuron during action potentials through voltage-gated calcium channels, binding to calmodulin and subsequently activating both calcium/calmodulin-dependent protein kinase (CaMK) II and CREB. On the other hand, calcium can also flow into the neuron during action potentials through NMDAr, activating the MAPK pathway. Although this is relatively slow and requires relatively high increases in calcium compared to the CaMK II or IV pathway, MAPK activity leads to strong and continuous neural activity. The binding of calcium to NMDAr activates Ras Protein-Specific Guanine Nucleotide Releasing Factor 1 (Ras-GRF1), a guanine nucleotide exchange factor (GEF) of the R-Ras family (a group of small GTPases) [41,42].

Ras-GRF in turn activates Ras, a small cell membrane-bound GTPase that plays a key role in the transduction of responses to various extracellular signals that affect

several cellular processes, mainly cell proliferation, differentiation, and apoptosis. Ras subsequently enables the activation of c-Raf (also known as Raf-1), a protein kinase that plays a critical role in intracellular signaling [41].

The c-Raf protein in turn allows for the activation of the MAPs pathways and their subsequent translation to MAPK, MAP2K, or/and MAP3K (depending on the cellular function to be triggered), which target the cell nucleus to activate three transcription factors or regulatory elements that bind to the promoter of the *c-fos* gene located in its 5′ untranslated region. For the fos promoter to bind to MAPs, there must be three regulatory elements attached to it. The first regulatory element is the serum response element (SRE), also called the dyad symmetry element (DSE). The SRE is located approximately 300 base pairs (bp) upstream of the transcriptional start site in the *c-fos* promoter and functions as an inducible enhancer element (in response to both protein kinase C-dependent and C-independent intracellular signaling pathways but not in response to the cAMP pathway) that binds, in turn, to the other two regulatory elements; one is the 67 kDa nuclear protein called serum response factor (SRF), which is expressed in most cells, and the other factor is the cAMP response element (CRE) that is present in several genes [3,15,41,43,44].

When the *c-fos* promoter is ready, the cAMP response element binding protein (CREB) is phosphorylated by ribosomal protein kinase S6 (rpS6) to bind to the CRE. Next, the phosphorylated ternary complex transcription factor (p62TCF or Elk-1), in association with SRF, binds to the *c-fos* promoter (SRE), initiating c-Fos protein synthesis at the nuclear level and subsequent translation at the cytoplasmic level. The Fos protein returns to the nucleus where it acts, together with c-Jun, as a third messenger by binding and forming the AP-1 heterodimer. Notably, this dimerization is necessary for DNA binding activity as it increases the stability of protein–DNA interactions. Fos is incapable of forming homodimers [3,15,35,43,45,46].

AP-1 interacts with DNA at two classes of cognate regulatory sequences: 12-O-tetradecanoylphorbol-12-acetate (TPA) (which affects the TRE response element) and cAMP (which affects the CRE response element) spanning the 5′-TGACTCA-3′ and 5′-TGACGTCA-3′ sequences, respectively. This binding is accomplished by a DNA domain called the basic leucine zipper (bZIP), which is composed of N-terminal DNA (DBD) and the C-terminal leucine zipper (LZ) [23,36–38].

bZIP is a highly conserved small nucleotide sequence (or protein motif). It localizes to a wide variety of cellular proteins that generally contain a distinct leucine at every seventh position within five successive heptads of amino acid residues, which facilitates hydrophobic interactions between the monomers of these proteins, the adoption of continuous $\alpha$-coiled helices in the Jun-Fos heterodimer, and insertion into major DNA grooves via strong van der Waals-type bonds. Several reports suggest that Jun and Fos bind to DNA as monomers and dimerize during association with DNA [1,22].

### 3.2.3.1. c-Fos Degradation

The *c-fos* product not only has a role in gene activation, but also acts as a negative regulator of its expression. This repressive function may be necessary for the rapid decrease in IEG expression after induction. The degradation process requires continuous protein synthesis. Dysregulation in the expression of IEGs such as c-Fos leads to abnormal cell proliferation in rats, and their inhibition blocks proliferation and cell cycle progression (which could be compensated by other members of the IEG family). These nuclear proteins are determinants in the proper control of cell proliferation. However, this does not mean that c-Fos (or other IEG products) cannot be re-induced, since in certain circumstances (e.g., in epileptic seizures) there is a refractory period for c-Fos re-induction to occur, which can last for several hours after the primary stimulus. This refractory period has been related to the action of Fra-1 (which is one of the four cellular Fos proteins mentioned above), as it causes transcriptional repression mediated by the *c-fos* promoter [15,47].

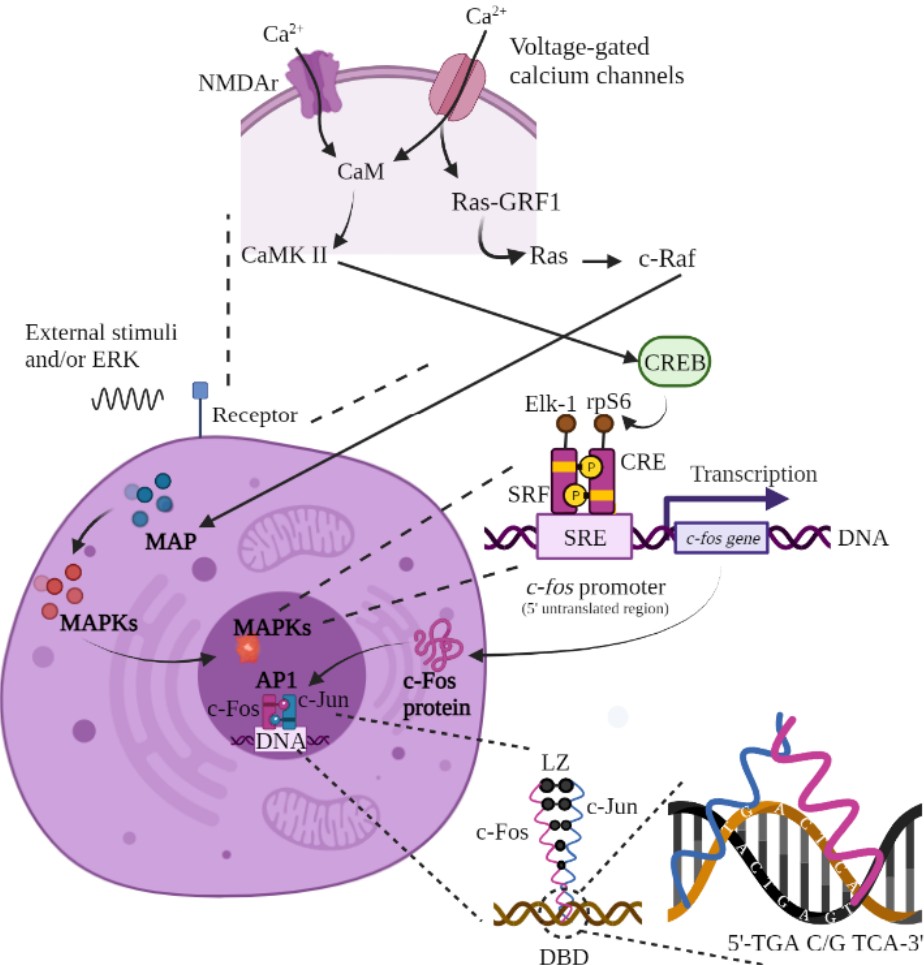

**Figure 1. Synthesis of c-Fos.** Calcium fluxes into the neuron during action potentials via NMDAr and voltage-dependent calcium channels. The influx of calcium binds calmodulin, activating CaMK II or IV and then CREB. NMDAr activity also activates RasGRF1, activating the MAPK (I, II, or III) pathway via Ras and c-Raf activation. The synthesis process for the Fos protein begins with an extracellular stimulus that activates the MAPK cascade and the *c-fos* gene through binding of the Elk-1, CRE, SRF, and rpS6 proteins to the SRE promoter. The active *c-fos* gene leads to translation of the c-Fos protein at the nucleus level and its subsequent transcription in the cytoplasm. Once created, c-Fos returns to the cell nucleus to bind to c-Jun and form the AP-1 heterodimer, which will be integrated into the DNA of the target gene (in the 5′TGA C/G TCA-3′ sequence) to trigger the desired answer. The basic leucine zipper allows for the union of c-Fos and c-Jun. Figure of original authorship, based on [3,15,22,23,35–38,41,43].

Proper regulation of the presence of c-Fos can be carried out by several mechanisms. One of them is the ubiquitin–proteasome system, in which ubiquitin is activated by the ubiquitin-activating enzyme (E1), thus transferring a ubiquitin transporter protein (E2) to a third enzyme (ubiquitin–protein ligase, E3) that catalyzes the covalent conjugation of ubiquitin to the protein substrate that binds to the ligase (in this case, c-Fos). After the formation of a multi-ubiquitin chain, the tagged protein is degraded by the 26S proteasome complex, thus releasing free and reusable ubiquitin to restart the process [46]. Other systems for passive proteolysis of Fos may involve SER, other IEGs, neurotransmitters, or proteins [e.g., EGTA or 2-aminopurine protein kinase (2AP)] [2]. To cite an example at the level of the cerebellum, Fos degradation could involve an interaction between parallel fibers (considering that these are granule cell axons) and Purkinje dendrites at the level of the molecular layer, where retrograde inhibition of neurotransmitter release in presynaptic parallel fibers occurs through the action of retrograde second messengers

(such as endocannabinoids) released by postsynaptic Purkinje dendrites, thus carrying this mechanism to the soma of granule cells and triggering proteolytic degradation of Fos. A third proteolytic system for c-Fos is produced by certain steroid hormones, but much remains unknown [1,48]. Times at which Fos degradation occurs are also not yet well defined, since it depends on whether it was either basal expression (based on its cellular function or brain area) or due to external stimuli, as mentioned above (see Table 3).

### 3.3. Fos Mapping outside the Central Nervous System

Outside the central nervous system (CNS), *c-fos* localization is usually reflected in bones (rib cage and vertebrae, including bone marrow), muscles, parts of the pleura, peritoneum subcutaneous tissue, and placenta (during the whole gestation period, in mice up to 18 days). This leads to a possible specific regulatory role of *c-fos* during the "differentiation-dependent growth" of fetal bone and mesodermal network tissue [4,25]. On the other hand, the localization and distribution of c-Fos leads to elevated levels in fetal liver and hematopoietic cells (mast cells, blood monocytes, differentiated neutrophils, and macrophages). Additionally, during mid-gestation, it originates and is expressed mainly in basal decidua of maternal origin or fetal cytotrophoblasts [18]. Outside the fetal stage, c-Fos protein has been detected in the lung tissue of young adult rats, mainly when there is acute lung injury [7].

Fos is also found in the skin, with a collagen degradation function that occurs through phosphorylation of JNK, c-Jun, and c-Fos due to UV-B radiation that promotes oxidative stress. The loss of collagen leads to a decrease in tensile strength and dermal stability [49].

### 3.4. Fos Activation in Glial Cells

It was previously believed that glial cells did not express *c-fos* basally, with expression only thought to occur after noxious stimuli.; however, it has recently been shown that this is not the case. In contrast to neurons whose Fos expression levels are usually associated with, but not limited to, depolarization, glial cells (astrocytes, microglia, and oligodendrocytes) do so mainly through the influence of proliferation, differentiation, growth, inflammation, repair, damage, plasticity, and other cellular conditions, providing additional information related to the brain microenvironment [18].

Concerning the expression of c-Fos after noxious stimuli in glial cells (microglia, astrocytes, and oligodendrocytes), its proliferation after brain injury leads to the secretion and release of neuronal growth factors (such as nerve axon regeneration factors) in the area surrounding the injury (and some surrounding areas) to enhance neuronal regeneration [1,20,33].

In monocytes-macrophages and astrocytes, inflammatory processes [such as those produced by lipopolysaccharides (LPS) and interleukins (IL) -1b)] could induce the expression of *c-fos* through the ERK and/or p38$^{MAPK}$ pathway and the subsequent phosphorylation of Elk1 (ETS like-1 protein) in the *c-fos* promoter at the SRE or CRE level [50].

Elevations in intracellular calcium ($Ca^{2+}_{int}$) are associated with an increase in the expression of *c-fos*. Astrocytes increase $Ca^{2+}_{int}$ levels in response to stimulation of glutamate receptors [quisqualate acid (QA) or AMPA and kainate (KA), but not NMDA] in two ways: (a) the entry of $Ca^{2+}_{ext}$ due to the opening of its voltage-sensitive channels; and (b) activation of the inositol phospholipid second messenger cascade, which results in the generation of inositol triphosphate and the release of $Ca^{2+}_{int}$. Levels of $Ca^{2+}_{int}$ are not only increased by activation of glutamate receptors, but also by high concentrations of potassium ($K^+$), which secondarily activates the voltage-gated $Ca^{2+}$ channels and allows for their entry and subsequent proliferation in the interior of the cell. High concentrations of $K^+$ (not less than 140 mM), together with the action of QA (mainly) and KA, induce the expression of *c-fos* in astrocytes. It should be mentioned that QA can induce *c-fos* by itself without needing the presence of $Ca^{2+}_{ext}$, contrary to the effect of $K^+$ which necessarily requires the presence of $Ca^{2+}_{ext}$ [51]. It is noteworthy that although glial cells express a

wide variety of glutamate receptors, they are more resistant to excitotoxicity-induced cell death than neurons, and they are more involved in tissue responses after injury [50].

Following seizure activity, there is an increase in c-Fos (but not *c-fos*) expression after 1 h, reaching a peak level after 12–24 h in both oligodendrocytes and astrocytes, as these proliferate in response to brain injury. Similarly, thymidine (a nucleoside with an important function during DNA synthesis and cell division) appears in its radioactive form (3H-thymidine) alongside high concentrations of c-Fos 72 h later, which is the time at which maximum cell division is known to occur. Therefore, the Fos protein could be related to the processes of cell division and differentiation after brain damage [20,32].

Inflammatory diseases at the CNS level, such as multiple sclerosis and Alzheimer's disease, also increase c-Fos expression at the spinal cord (SC) level in astrocytes as they become reactive in order to reduce inflammatory processes [33]. Following a mechanical injury to the cerebral cortex, c-Fos activation exhibits peak levels in oligodendrocytes and astrocytes between 12 and 24 h after brain damage [20].

### 3.5. Stimuli That Trigger Fos Expression in Mammalian Animal Models

Basal levels of *c-fos* and its product (c-Fos) are very low but detectable in some adult mammalian neurons. Higher levels of *c-fos* and c-Fos are usually transiently expressed in certain brain regions after certain stimuli [16,20]. Immunohistochemistry is generally used to detect the presence of c-Fos, Northern blot analysis is carried out for the expression of *c-fos*, and PCR analysis is used to measure levels of *c-fos* $_m$RNA.

The depolarizing action of glutamate leads to an increase in the $_m$RNA of this *c-fos* through activation of NMDA and AMPA receptors in various brain structures, such as the vestibular nuclei (by activation of the vestibulo-sympathetic reflex that modulates blood pressure during postural changes) [52]. Dopamine also has the same effect (through its D1 receptor at the level of the striatum in animal models with Parkinson's disease) [53]. Likewise, norepinephrine and epinephrine increase *c-fos* expression by binding directly to its $\alpha$2 adrenergic receptors in neurons with autonomic functions or integrating somatosensory and/or affective functions with autonomic mechanisms in the locus coeruleus, central nucleus of the amygdala, bed nucleus of the stria terminalis, nucleus of the solitary tract, ventrolateral medulla oblongata, paraventricular nucleus, and supraoptic nucleus [23]. When acetylcholine binds to its nicotinic receptors on neurons (mainly non-dopaminergic), it increases the expression of c-Fos in the medial terminal nucleus of the accessory optic system, the superior colliculus, the interpeduncular nucleus, and the caudal linear subnucleus of the ventral tegmental area, and can also increase expression in its muscarinic receptors at the level of the cortex [19]. Serotonin activates *c-fos* at the level of the dorsal striatum, the paraventricular nucleus of the hypothalamus (PVN), and the central amygdaloid nucleus in obese animal models [54].

Ions play an important role in the positive regulation of *c-fos*. As in glia, high concentrations of $K^+$ and $Ca^{2+}$ stimulate their activation. During cell stimulation, $Ca^{2+}_{int}$ is increased by the action of NMDA receptors, inducing *c-fos* $_m$RNA and its consequent protein. The $Ca^{2+}$ increase, in turn, can activate multiple kinase pathways, including MAPK, CaMK II and IV, and cAMP-dependent protein kinase A (PKA), which can terminate in Fos activation [1,3,42].

Some physiological mechanisms also lead to an increase in *c-fos*. Massive $Ca^{2+}$ influx promotes cell death, inhibits protein synthesis, and induces gene $_m$RNA upregulation. Regarding the c-Fos protein, cell division or signals associated with neuronal excitation have the same effect by activating voltage-dependent $Ca^{2+}$ channels. Oxygen free radicals suppress *c-fos* expression, although the mechanism has not yet been fully elucidated. On the other hand, for c-Fos, the aging process increases expression in mice and dogs at the SC level due to neuronal distribution and natural neuronal damage/death caused by old age [4,14].

Following a mechanical injury to the cerebral cortex, *c-fos* activation appears at the neuronal level up to 72 h later, with peak levels between 6 and 12 h after brain damage. It is

worth mentioning that the time of visualization of *c-fos* expression may also depend on the type of antibodies used, since it has been observed that the gene can be expressed up to 72 h after injury when polyclonal antibodies are used but ceases to be expressed 4 h after injury when monoclonal antibodies are used [20]. Likewise, spinal cord injury (SCI) also increases c-Fos expression 2, 6, 12, and 24 h after injury, extending to supraspinal areas such as the cerebellum (vermis and hemispheres), hypothalamus (supraoptic and paraventricular nuclei), and brainstem nuclei (medullary reticular formation, area postrema, nucleus of the tractus solitarius, inferior olive, prebötzinger complex) [30,31,55].

In ischemic environments, *c-fos* can be induced at the level of the dentate gyrus 15 to 60 min after bilateral occlusion of the common carotid artery in gerbils, and 4 h later in the hippocampus (returning to baseline after 6 h). This expression is due to membrane depolarization caused by disruption of ion pumps, $Ca^{2+}$ influx, uncontrolled neurotransmitter release, and altered cell membrane structure. However, the involvement of glia must also be considered, as *c-fos* activation signals the passage of cells from the resting state to the division phase [24].

Seizure induction by pentylenetetrazol, kainic acid, or metrazol in rodents and marmosets showed massive increases in c-Fos expression at the level of the cingulum, piriform, and motor cortex after 30–60 min, reaching peak expression levels 1–2 h after seizures in rodents and after up to 6 h in marmosets; however, in the hippocampus, peak levels were observed 30–60 min later, returning to baseline after 24 h. In the case of *c-fos*, its expression is induced from 30 to 90 min, reaching its peak levels after 3–6 h in the same regions and returning to baseline after 3 h (or 3–4 h in pyramidal cells) while remaining present in the brain up to 17 h after the seizure [1,34].

Alcohol (ethanol) exposure in rats promotes the upregulation of this gene in the pyriform cortex and hippocampus after 2–4 h, reaching its maximum expression at around 8 h [1]. Temperature changes also activate its expression. Constant (1 h) and high exposure to heat significantly increases (compared to cold) the concentration of this gene at the level of the neuronal nuclei of the hypothalamic preoptic area, which are involved in thermoregulation [35].

Repeated stimuli modify the basal concentration of *c-fos*. In rats with hippocampi sensitized by constant seizures, an increase in NGFI-A gene expression was observed at 5 min, an increase in *c-fos* at 30 min, and an increase in *c-jun* at 30 min. Although *c-fos* returns to baseline after 30 min, *c-jun* and NGFI-A continue to increase after 8 h. Similarly, constant cocaine injections increase *c-fos* signaling. In contrast, repetition of negative emotions such as depression can decrease c-Fos expression in the prefrontal cortex for up to 7 days after stimulus presentation [3,9].

Water deprivation also alters *c-fos* expression. It has been observed that when hypertonic solutions (HS) are administered for 24 h, an increase in this gene is generated after 30 min at the level of the supraoptic nuclei (SON) and PVN of the hypothalamus, ventrolateral medulla oblongata, nucleus of the tractus solitarius, and locus coeruleus, reaching its peak after 30–60 min and returning to baseline after either 4–8 h or up to 24 h later [1].

Noxious stimulation (e.g., by administration of formalin, acetic acid, capsaicin, and/or scorpion venom or via tissue injury) also induces *c-fos* activation in the neurons of lamina I (mainly), II, V, and VI of the SC and, to a lesser extent, VII, VIII, and X of the hind limbs of rats after 1–2 h. Expression of c-Fos is induced after 30–60 min in dorsal horn neurons [3,36].

Light and dark periods also influence the expression of this IEG. Light/dark (L/D) cycles synchronize circadian rhythms by altering neuronal activity in the suprachiasmatic nucleus (SCN) of the hypothalamus, providing a circadian pacemaker that receives direct (retinohypothalamic pathway) or indirect (retino–geniculate hypothalamic pathway) information from the retina. Exposure to light during the dark phase of the L/D cycle causes an increase in *c-fos* in the SCN, intergeniculate follicle (IGL), and ventral lateral geniculate nucleus (vLGN), but not during the light phase. The pineal gland, hippocampus,

putamen, and caudate nucleus also express an increase in *c-fos* after the onset of the dark cycle, although their basal levels are almost nonexistent [1].

Learning and memory processes require protein synthesis to develop long-term changes in the CNS. Both *c-fos* and c-Fos increase their expression during long-term potentiation (LTP) in chicks (non-mammalian animal) at the level of the cerebral cortex and in rats learning to discern/avoid certain stimuli or actions at the level of the hippocampus. However, when LTP has been established, an increase in these IEGs (at the level of the dentate gyrus) is not observed, though others, such as *c-jun*, Jun-B, and Jun-D are increased; thus, *c-fos* possibly does not intervene in the memory process [18].

Stress is another mechanism that contributes to the normal functioning of the organism. Inducing stress in animals through various stimuli (such as colchicine or HS administration, immobilization or capsaicin treatment, induction of mild shocks or leg shaking, conditioned and unconditioned fear, handling for the first time, or exposure to tones) results in an increase in *c-fos* $_m$RNA in the PVN (an area that secretes the corticotropin-releasing factor or expresses glucocorticoid receptors), as well as in structures that synthesize catecholamines, i.e., the ventrolateral medulla oblongata, the nucleus of the tractus solitarius, the amygdala, and the locus coeruleus [16].

### 3.6. Stimuli That Trigger Fos Expression in Non-Mammalian Animal Models

The majority of studies on Fos (gene and protein) are based on the use of mammalian animals, possibly due to the great understanding and standardization of their use. However, this does not mean that Fos is limited only to this type of model (Table 4).

**Table 4.** Fos-triggering stimuli and their animal models.

| Stimuli | Model |
| --- | --- |
| Neurotransmitters | |
| Glutamate depolarization (NMDA and AMPA) | Rats/blood pressure and posture |
| Dopamine (D1) | Rats/Parkinson's |
| Noradrenaline and adrenaline ($\alpha$2) | Rats/neuroplasticity |
| Acetylcholine (nicotinics) | Rats/visual stimuli |
| Serotonine (5-HT$_{1c}$ and 5-HT$_2$) | Rats/obesity |
| Physiological mechanisms | |
| Alzheimer's | Rats, humans |
| Mechanical brain injuries | Rats (neurons: 1, 6, 12, and 72 h later; glia: 12, 24, and 72 h later *) |
| Ischemia | Mongolian gerbils, rats (15–60 min, 3 days *) |
| Heatstroke | Rats, rabbits |
| Seizures | Rats (30–60 min, 3–4 h *), marmosets (6 h *) |
| Learning and memory | Rats, mice |
| Osmotic stimulation | Rats (30–60 min, 180 min, 1–2 h, 4 h *) |
| Stress | Rats |
| Cardiac rhythms | Rats, hamsters |
| Alcohol intake | Mice, rats (2–4 h, 8 h *) |
| Depression | Mice (7 d) |
| Old age (neuropathic pain) | Beagles, mice |

[1,14,19,23,34,52–54]

* Depending on the brain area analyzed and the animal model used.

It has been observed that c-Fos increases its expression at the level of the hypothalamic arcuate nucleus after ingestion of ferulic acid (FA). This is because FA has physiological benefits as an antioxidant and stimulates appetite in birds [chicks, broilers (*Gallus gallus*)] [56]. Meanwhile, it has also been observed that c-Fos could be involved in the expression of osteoclastogenesis regulatory factors and in the regulation of bone mineral density in Atlantic cod (*Gadus morhua*), inhibiting osteoclast formation and bone resorption through a mechanism that is still unclear [57]. In larval and adult zebrafish, c-Fos protein may also increase due to augmented trypsin, which is released from the gills of freshwater and saltwater fish upon stress or injury, serving as an escape response to certain threats (temperature, pH, overcrowding, and injury) [58]. Finally, there are few studies on c-Fos expression in reptiles. One of the most recent analyzed fear behavior, specifically freezing in lizards (*Tropidurus hispidus*). When these animals were subjected to the presence of intimidating stimuli (such as the presence of felines), Fos expression increased in telencephalic areas such as the medial cortex, dorsomedial cortex, lateral cortex, amygdala, striatum, septum, and dorsal posterior ventricular region, thus demonstrating that Fos is linked to the fear response in these animal species [59].

## 4. Perspectives and Suggestions

During this review, it has been shown that Fos (both as a protein and as a gene) is widely used as a marker of cellular activity. Despite this, there remains much controversy regarding its use since, as an IHC technique, it has several disadvantages and limitations: (a) It requires careful controls to eliminate extraneous and not always obvious sources, e.g., anesthesia, stress, circadian rhythms, light, sound, etc. With sentient animals, even seemingly mild forms of manipulation lead to Fos expression in the brain, giving the results a false-positive feel. (b) Although expressed in many systems in response to stimulation, IEGs are not universal markers of neuronal activity as they can also be present in glial cells, making it impossible to identify which cell type is active. (c) Since c-Fos is an indicator of cellular activity, neurons that are inhibited will not express this protein, limiting the study of excitatory pathways. (d) The lifetime of the protein to quantify is short, so this technique is only useful for experiments whose stimuli are fast and short in duration. (e) Although Fos IHC reveals possible active synaptic pathways after a stimulus, it does not differentiate whether it is polysynaptic or multisynaptic; therefore, the results must be interpreted while taking this limitation into account. (f) With Fos IHC, it is not possible to separate the neuronal activities present (e.g., emotional and cognitive, motor, sensory, autonomic, etc.). (g) The presence of Fos does not provide information on which downstream pathways are also activated by the stimulus.

However, although the use of Fos has many limitations, it also has great advantages that keep it in use within neuroscience: (a) it is expressed in a wide range of harmful and non-harmful stimuli, so it can be used for almost any response one wants to analyze; (b) it is also useful in a wide range of animal models, allowing for lower costs, easy handling, and reproduction of the same study; (c) Fos IHC is specific, fast, and powerful; (d) it is relatively easy to perform and quantify; (e) Fos protein is expressed in the nucleus and cytoplasm, so cellular mechanisms can be analyzed in almost all target cells; and (f) IHC can be used in combination with various quantitative techniques, allowing several response parameters to be analyzed without them interfering with each other. As an example for this last section, if cellular activity is to be analyzed, IHC-Fos would be used, but if neuronal degeneration is also to be detected, the fluoro-jade B (F-J B) technique would be incorporated. On the other hand, if the objective is to confirm or quantify changes in the expression levels of certain proteins, a Western blot (WB) would be added. Similarly, Fos-IHC can be complemented with quantification techniques of neuronal nuclei by NeuN-IHC or with glial detection techniques incorporating GFAP (glial fibrillary acidic protein).

## 5. Conclusions

The activation of *c-fos* and c-Fos in the brain is induced by various stimuli and used as a tool to study cell activation and external cellular signals with phenotypic changes in nerve cells. The level and timing of gene expression, as well as its product, varies according to the stimulus and the area studied, as well as the mechanisms of action. The exact role of Fos in the nervous system is still unclear because, as discussed during this article, there are many factors that trigger it. Although Fos has been studied in several experimental animal models since the 1990s, its use as a marker of cellular activation for many physiological processes makes it still useful and reliable.

**Funding:** This research received no external funding.

**Institutional Review Board Statement:** This paper was approved by the Institutional Review Board (IRB) of Universidad Veracruzana and Instituto de Investigaciones Cerebrales.

**Informed Consent Statement:** Informed consent has been obtained from all individuals included in this article.

**Data Availability Statement:** The data that support the findings of this article are cited in the part of "Methodology: Selection Criteria".

**Acknowledgments:** We are grateful to the Consejo Nacional de Ciencia y Tecnología (CONACyT) for the scholarship granted to S.Y.L.A. (Financing CONACyT-893213 doctoral fellowship).

**Conflicts of Interest:** The authors of this review article have contributed to its content and the development of the format of this manuscript. All authors declare that they have no conflict of interest in its publication.

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
