# Peer review of "Current Opinion on the Use of c-Fos in Neuroscience"

_neurosci, doi:10.3390/neurosci3040050_

Round 1

Reviewer 1 Report

The manuscript is devoted to the role of c-Fos in neurons and glial cells metabolism. The manuscript describes c-fos gene composition and transcription factors, fos-protein composition, activation pathways and degradation. Tle last part of the manuscript is devoted to use of mammalian and non-mammalian models for the investigation of c-Fos expression triggering. The manuscript is well-organized and can be accepted after minor revision due to correct some mistakes as, for example, "This protein it has a half-life of approximately 2 h".

Reviewer 2 Report

Comments and Suggestions for Authors

Based on the article’s title, I expected to find this article relevant and promising.  I did find it informative; however, the title does not reflect the article’s content. I would encourage the authors to consider shortening the article and changing its focus. Below are my specific observations and recommendations.

Overall, I find the literature that is presented to be incomplete, there are authors such as Gabriela González-Mariscal or Kurt Hoffman that have published interesting and relevant results on rabbit behavior and c-Fos expression in the nervous system, but that are not cited. I suggest that the authors expand the section of Fos- triggering stimuli and their animal models, giving more specific examples.  I believe that the authors neglected to include important and relevant literature here.  In the keywords list, the authors should include the term “animal models”.

With regards to the bibliography, the authors fail to cite some important work: for example, Curran and Morgan, 1986, and 1994, are both excellent reviews about c-Fos. There are studies of animal models of anxiety and studies of rabbit maternal behavior that could be included and should have appeared in the literature search using the search terms mentioned before.

In the document there are paragraphs that present an excessive number of references, comprising more than half of the paragraph’s content. In such cases, I suggest to include only the most relevant references.

On the following lines I will expand on my comments:

Line 10: check grammar

Line 12: change “countless”

Line 22: change “this reasons” – plural

Line 23: is your objective to know?

Line 29: I suggest that the first reference be of an English article, if you want to have a broader audience, probably you can check Morgan and Curran, 1986.

Line 34: repeated reference, there are many original works about c-Fos.

Line 47: It seems that if the term C-Fos is established in the title, then this same term should be used throughout the text.

Line 50: Repetition of word in the same line: “searched. The search”

The article of Herrera & Robertson was cited excessively throughout the article.

Line 95: you need a reference.

Line 96: Look for original reference of Stehelin, 1976, instead of Meza, 2006.

Table 3: Put the references where they correspond. Correct the accent of área: area. I don´t understand: ** depending to.

Line 158: There is an excessive number of references for one paragraph.

Line 174: I have counted around 7 times that Herrera & Robertson is cited within the review, although it is an interesting article, it seems that the reader better go to that reference.

Line 196: It is a long paragraph, with much information, but only has 1 reference.

Line 284: put the reference of the example that is mentioned.

Line 487: The original reference was translated from the abstract of the dissertation, there must be original references from the Dos Santos thesis. One would need to know Portuguese in order to understand the entire document, only the abstract is in English.

About the sections and tables

I would suggest shortening section 3, specifically sections 3.2.3 and 3.2.4. I don´t like how section 3.2.3 is written, the paragraph text is extensive and references are placed only at the end of the paragraph; the reader cannot associate a single reference with a single finding.  Please include a table for section 3.5

Section 3.3 Herrera & Robertson, 1996 is again cited. It seems suspicious that, considering the size of the bibliography, that those authors appear on every page (in page 12, this study is cited 6 times), even more than twice per paragraph (line 301, line 306), while  other references appear only one time in the whole review.

Section 3.3 can be expanded.

Section 3.5 would benefit from a table that summarizes this information.

Section 4. Perspectives and suggestions should be one of the more extensive sections of the review, considering the title (“Current opinion on the Use of c-Fos in Neuroscience”).

In Table 1 -in part I and II-, each row should include a reference relevant to the information that is presented in that row.

In the references of table 1, I would like to know more about the reference of Lea Silver, 2003. Do you have the book or the dissertation? I could not find it; if it is unavailable or not easily available to the reader, I suggest removing it.

Table 2. I find this table confusing; I suggest having a classical table design, the colors are not effective and in fact make the table difficult to understand. Also, I find difficult to understand to which animal model corresponds to each IEGs. The table would be more interesting if you could very briefly describe the experiment that was performed.

Table 4: include references according to stimuli, and be more precise in the specifics of the model. Use a classical table format. Include the relevant reference for each row (for each study summarized) within the row to which it corresponds.

Figure 1: Is this an original illustration by the authors, or is it an adaptation of a previously published illustration?

In general, the article needs a more extensive discussion of the authors’ opinion on the use of c-Fos in neuroscience: incorporating personal experience or work that the authors may have carried out, and what could be improved in the general science. Additionally, the authors need to work on the literature cited: in some areas of the text the cited literature is excessive, in other areas it is lacking.

Reviewer 3 Report

The review is easy to read and quite interesting. The review uses a lot of literature from the past 30 years.

The disadvantages of using c-fos/c-Fos as markers of neuronal activation have been extensively described in the work.

An interesting selection of data is presented in Table 2 and 3.

The section describing the stimuli that trigger Fos expression in mammalian animal models is well described.

Overall, the work deserves attention and publication in the appropriate journal.

I have some comments on the work below:

Line 39. Clarify about the brown nuclear label.

Line 108. These normal cellular sequences from which v-fos was derived were termed the fos or c-fos proto-oncogene, which is located at the level of the cell nucleus. What exactly is meant by which is located at the level of the cell nucleus?

Table 3. Expression time of c-Fos and c-fos: basal and after stimuli. c-fos should be in italics?

Check again the correct spelling of genes and their products throughout the review.

Line 188. N-methyl-D-aspartate glutamine receptor (NDMAr) or a glutamate receptor?

The regulation of c-Fos presence can be controlled by the process of mitophagy?

Line 363. Immunohistochemistry is generally used to detect the presence of c-Fos and Northern blot analysis is carried out for expression of c-fos. Is the PCR assay used to detect c-fos dynamics?

Round 2

Reviewer 2 Report

I reccommend the publication of the article.